# Risk factors and biomarkers of non-alcoholic fatty liver disease: an observational cross-sectional population survey

Xiao-Yu Hu,[1] Yun Li,[1] Long-Quan Li,[1] Yuan Zheng,[1] Jia-Hong Lv,[1] Shu-Chun Huang,[1] Weinong Zhang,[1] Liang Liu,[1] Ling Zhao,[1] Zhuiguo Liu,[1] Xiu-Ju Zhao[1,2,3]

[1]School of Biology and Pharmaceutical Engineering, Wuhan Polytechnic University, Wuhan, China
[2]Hubei Key Laboratory of Lipid Chemistry and Nutrition of Oil, Oil Crops Research Institute of the Chinese Academy of Agricultural Sciences, Wuhan, China
[3]Department of Nutrition and Food Science, Texas A&M University, College Station, Texas, USA

**Correspondence to**
Dr Xiu-Ju Zhao;
dzrdez@163.com

## ABSTRACT

**Objective** Non-alcoholic fatty liver disease (NAFLD) is a major public health burden in China, and its prevalence is increasing. This study aimed to determine the risk factors and biomarkers of NAFLD.

**Design** An observational cross-sectional primary survey.

**Setting** Central China.

**Participants** The study included 1479 participants aged over 18 and below 80 years, not currently being treated for cancer or infectious disease or no surgery in the previous year, and no history of cancer or an infectious disease. Participants underwent clinical examination, metabolomic assay and anthropometric assessment. Univariate and logistic regression analyses were used to evaluate associations between covariates and NAFLD.

**Main outcome measures** Risk factors and metabolic biomarkers including sex, body mass index, hypertension, body fat ratio, blood triglycerides, blood fasting glucose, liver enzyme elevation, uric acid and oleic acid-hydroxy oleic acid (OAHOA).

**Results** Data from the 447 participants (mean age 44.3±11.9 years) were analysed, and the prevalence of NAFLD was 24.7%. Male sex (OR 3.484, 95% CI 2.028 to 5.988), body mass index ≥24 kg/m$^2$ (OR 8.494, 95% CI 5.581 to 12.928), body fat ratio (≥25 for women, ≥20 for men) (OR 1.833, 95% CI 1.286 to 2.756), triglycerides ≥1.7 mmol/L (OR 1.340, 95% CI 1.006 to 1.785), fasting glucose ≥6.1 mmol/L (OR 3.324, 95% CI 1.888 to 5.850), blood pressure ≥140/90 mm Hg or antihypertensive drug treatment (OR 1.451, 95% CI 1.069 to 1.970), uric acid (≥357 µmol/L for women, ≥416 µmol/L for men) (OR 2.755, 95% CI 2.009 to 3.778) and OAHOA (<5 nmol/L) (OR 1.340, 95% CI 1.006 to 1.785) were independent predictors of NAFLD (all P<0.05). These results were verified by all 1479 participants.

**Conclusions** NAFLD was common among the study participants. In particular, NAFLD was correlated with uric acid. We identified OAHOA as a novel marker of NAFLD prevalence. It provides a reference on the prevention of NAFLD and related metabolic diseases with the rapid urbanisation, technological advancement and population ageing in China over the recent decades.

### Strengths and limitations of this study

► Risk factors of non-alcoholic fatty liver disease (NAFLD) in central China.
► Biomarkers of NAFLD using metabolomics.
► Prediction ability of metabolic markers using receiver operating characteristic curve.
► No information on education.

## INTRODUCTION

China has the world's largest population and is undergoing rapid economic growth and social reform. This advancement has paralleled demographic, lifestyle and cultural changes, which have exerted notable effects on the health profile of China's residents and placed significant constraints on the country's healthcare system.[1]

Such changes are apparent in major cities of central China, such as Wuhan. The prevalence of non-alcoholic fatty liver disease (NAFLD) is increasing, now the most common chronic disease among Chinese adults in China, and the rates exceed those of equally important epidemics, including obesity, hypertension and type 2 diabetes. NAFLD is followed by an increase in overall cardiovascular morbidity and mortality.[2–4] It has been estimated that NAFLD will be the most frequent indicator for liver transplantation and regeneration in the coming decades.[5–7] Additionally, because population ageing and the continual increase in obesity and hypertension rates over the recent decades, the prevalence and impact of NAFLD in China are expected to increase. Accordingly, NAFLD is a major public health burden.

The reported prevalence of NAFLD among Chinese adults ranges from 15% to 30%.[5–7] NAFLD prevalence increases with age, most notably from the fourth decade of life onwards (40–60 years of age).[5 6 8 9] Because of

endocrine function variations and fat redistribution, the prevalence and risk factors for NAFLD may vary across different ages and between male and female populations.[10–13] However, the association of age and sex with NAFLD in central China is still unclear. Furthermore, the association of hypertension, obesity, hyperlipidaemia, insulin resistance and diabetes with NAFLD has been widely evaluated in adult cohorts,[14–16] and these metabolic traits are well-accepted risk factors at present for NAFLD. Nonetheless, limited data are available on the association of NAFLD with other covariant traits.[17 18] Metabolomics provides integrated information on biological status and can investigate molecular variations with disease phenotyping[19 20]; several markers for NAFLD have been identified previously.[5 6] Sphingolipids (SLs) and branched fatty acid esters of hydroxy fatty acids (FAHFAs) have been reported to protect against autoimmune and allergic disorders and type 2 diabetes, respectively[21 22]; the effects of SLs and FAHFAs on NAFLD are largely unclear.

Our present study was aimed at investigating the prevalence of NAFLD, elucidating the association of NAFLD with age, sex, metabolic factors and markers of liver injury, and identifying novel metabolites associated with NAFLD risk, via clinical examination, anthropometric assessment and targeted metabolomics.

## METHODS
### Study population
All participants provided written informed consent prior to enrolment. The Wuhan study is an observational cross-sectional study of participants recruited from the Wuhan municipality in central China. Participants were recruited from routine health check-ups to assess the utility of health check-ups in determining the prevalence of and risk factors for NAFLD. Each participant completed a comprehensive interview and clinical examination that involved the collection of fasting blood and urine samples, abdominal ultrasonography, and anthropometric assessment.

### Interview
The interview, followed by the clinical examination, was performed by a physician and was designed to obtain information on demographic characteristics, medical history and comorbid conditions. Participants aged over 18 and below 80 years were included if they were not currently being treated for cancer or infectious disease or had undergone surgery in the previous year, and if they had no history of cancer or an infectious disease.

### Biochemistry
Fasting blood and urine samples were collected on clinical examination. Blood alanine aminotransferase (ALT), alkaline phosphatase, aspartate aminotransferase (AST), gamma-glutamyl transferase, glucose, total bilirubin, triglycerides and uric acid levels were measured using automatic enzymatic procedures (Nanjing Jiancheng Sci-tech, China). Hepatitis B surface antigen (HBsAg),

triiodothyronine, thyroxine (T4) and thyroid-stimulating hormone were examined via automatic immunoassays (Roche Diagnostics, Mannheim, Germany).

### Diagnosis of NAFLD
Abdominal ultrasonography was performed on all participants by accredited medical technicians using a Hitachi HI VISION 900 ultrasound machine. Images were evaluated by a senior pathologist. NAFLD was diagnosed by the technician according to ultrasonography.[23–25] Participants with any of the following possible secondary causes of fatty liver were excluded from the current analyses: (1) excessive alcohol intake,[26] (2) anti-hepatitis C virus (HCV) or positive HBsAg, or (3) use of drugs historically for treating fatty liver (ie, amiodarone, corticosteroids, methotrexate or tamoxifen).[13]

### Metabolic covariates
Anthropometric measurements were performed by accredited nurses. Body mass index (BMI) was defined as the measured weight (kg) divided by height squared ($m^2$). Body fat ratio (BFR) was equal to fat weight (kg) divided by body weight (kg). The average of two resting blood pressure (BP) measurements at a single visit after a 10 min break was used for analysis.

Metabolic traits were defined as the following: obesity (BMI ≥28 kg/$m^2$), hypertension (BP ≥140/90 mm Hg or antihypertensive drug treatment), BFR ≥25 for women or ≥20 for men, blood triglycerides ≥1.7 mmol/L, blood fasting glucose ≥6.1 mmol/L, liver enzyme elevation (AST ≥40 U/L or ALT ≥40 U/L), uric acid ≥357 μmol/L for women or ≥416 μmol/L for men, or impaired oleic acid-hydroxy oleic acid (OAHOA) (<5 nmol/L).

### Targeted Liquid Chromatography/Mass Spectrometry analysis of SLs and FAHFAs
To identify SLs,[21] silica column-affectionate lipid fractions were analysed by reverse-phase high-performance liquid chromatography-tandem mass spectrometry (HPLC-MS)/MS (Agilent C18 column connected with Thermo Scientific LTQXL; gradients of water to methanol, 10:90 to 0:100) to identify spots present in both control and liver disease. Identified lipid species of interest were purified by reverse-phase HPLC (Varian Prostar/Agilent C18 column) and subjected to structural analysis.

FAHFAs were measured by HPLC-MS[22] (an Agilent 6410 Triple Quad liquid chromatography (LC)/mass spectrometry (MS) via multiple reaction monitoring (MRM) in a negative ionisation mode). Briefly, extracted and fractionated samples were dissolved in 25 mL MeOH; 10 mL was injected for further analysis. A Luna C18(2) (Phenomenex) column was used with an in-line filter (Phenomenex). Distinct FAHFAs were dissolved via isocratic flow (0.2 mL/min for 120 min, solvent: 93:7 MeOH:$H_2O$, 5 mM ammonium acetate, 0.01% ammonium hydroxide). Transitions for OAHOAs were m/z 561.5 → m/z 281.2 (collision energy (CE)=30 V), and m/z 561.5 → m/z 279.2 (CE=25 V).

**Table 1** Participant characteristics

| Indicators | Total | NAFLD | No NAFLD | P value |
|---|---|---|---|---|
| **(A) Subset** | | | | |
| n | 447 | 102 (22.8%) | 345 (77.2%) | |
| Age (years) | 44.3±11.9 | 44.6±10.8 | 44.0±12.1 | 0.609 |
| Age range | 4.0±1.2 | 4.0±1.2 | 3.9±1.2 | 0.608 |
| Obesity | 235 (51.8%) | 94 | 141 | <0.001 |
| Body fat | 309 (68.1%) | 89 | 220 | <0.001 |
| Blood lipids | 87 (19.2%) | 21 | 66 | 0.745 |
| Hypertension | 55 (12.1%) | 23 | 32 | 0.003 |
| Male | 171 (38.3%) | 64 | 107 | <0.001 |
| Elevated liver enzymes | 18 (4.0%) | 7 | 11 | 0.174 |
| **(B) Total study population** | | | | |
| n | 1479 | 365 (24.7%) | 1114 (75.3%) | |
| Obesity | 809 (54.7%) | 334 | 475 | <0.001 |
| Body fat | 1003 (67.8%) | 309 | 694 | <0.001 |
| Blood lipid | 458 (30.9%) | 163 | 295 | <0.001 |
| High blood glucose | 71 (4.8%) | 43 | 28 | <0.001 |
| Hypertension | 314 (21.2%) | 129 | 185 | <0.001 |
| Abnormal uric acid | 83 (5.6%) | 38 | 45 | <0.001 |
| Male | 590 (39.8%) | 214 | 376 | <0.001 |
| Impaired liver function | 32 (0.2%) | 12 | 20 | 0.142 |
| Elevated liver enzymes | 110 (0.7%) | 53 | 57 | <0.001 |

Age ranges of 20–30, 30–40, 40–50, 50–60 and ≥60 are 2, 3, 4, 5 and 6.

Conditions were defined as follows: obesity (body mass index ≥24 kg/m$^2$), hypertension (blood pressure ≥140/90 mm Hg or antihypertensive drug treatment), body fat ratio ≥25 for women or ≥20 for men, blood triglycerides ≥1.7 mmol/L, blood fasting glucose ≥5.6 mmol/L, impaired liver function (positive HBsAg), liver enzyme elevation (AST ≥40 U/L or ALT ≥40 U/L), and uric acid ≥357 μmol/L for women or ≥416 μmol/L for men.

Mean values are provided with SD, unless otherwise noted as n (%). Differences between participants with and without NAFLD were evaluated with t-tests or the Wilcoxon-Mann-Whitney test for continuous variables and the $\chi^2$ test for categorical variables.

ALT, alanine aminotransferase; AST, aspartate aminotransferase; HBsAg, hepatitis B surface antigen; NAFLD, non-alcoholic fatty liver disease.

## Statistical analyses

Baseline characteristics were analysed using descriptive statistics. U tests, $X^2$ tests or Wilcoxon rank-sum tests (for medians) and Student's t-tests (for means) were used to evaluate the significance of differences in the distribution of categorical data and continuous data, respectively. To examine the relations between traits and NAFLD, we performed binary or multiple logistic regression analyses. We calculated the area under the receiver operating curve to assess prediction ability of metabolic markers. A P value of <0.05 was considered statistically significant unless stated otherwise. Data analyses were performed using SPSS V.17.0.

## RESULTS
### Study population

In total, 1500 participants were enrolled. Twenty-one participants were excluded due to absence of clinical data. Thus, a total of 1479 study participants were included in the final analysis, of whom 447 reported their age and had no more than one disease, if any (denoted as the 'subset'). Patients' general characteristics are shown in table 1. Men accounted for 40% of the study population (38% of the subset). Participants' mean age was 44.3±11.9 (range 20–74) years.

### Prevalence of NAFLD

The prevalence of NAFLD, as determined by ultrasound and biopsy, was 24.7% (22.8% in the subset).

The peak prevalence of NAFLD was 26.4% and 26.3% in this study between the ages of 30–40 years and 50–60 years, respectively (table 2).

The prevalence of NAFLD demonstrated an increasing trend with advancing age (OR 1.049, P=0.607), and after adjustment for sex and metabolic features NAFLD tended to be inversely correlated with age (OR 0.844, 95% CI 0.667 to 1.068; P=0.157). NAFLD was diagnosed in 37.4% of men and 13.8% of women (P=0.084). In multivariate analysis, after adjustment for metabolic features and age, male sex was associated with NAFLD (OR 3.484, 95% CI 2.028 to 5.988; P<0.001; table 3A).

**Table 2** NAFLD prevalence across age range

| Age range (years) | Participants, n | Participants with NAFLD, n | NAFLD prevalence,% | U test |
|---|---|---|---|---|
| 20–30 | 64 | 10 | 15.6 | 1.22 |
| 30–40 | 102 | 27 | 26.4 | 0.75 |
| 40–50 | 120 | 26 | 21.7 | 0.18 |
| 50–60 | 111 | 30 | 27.0 | 1.00 |
| ≥60 | 50 | 9 | 18.0 | 0.64 |
| Total | 447 | 102 | 22.8 | 1.64 (α=0.1) |

NAFLD, non-alcoholic fatty liver disease.

## Association between NAFLD and metabolic features

Among metabolic covariates, obesity and hypertension occurred more frequently in participants with advancing age (OR 1.471 and 1.822, respectively; P<0.001). Increased body fat was more prevalent in women than in men (OR 2.042, P<0.001 (OR 1.754, P=0.007 in the subset)), and obesity (OR 0.537, P<0.001), increased blood lipids (OR 0.829, P<0.101) and hypertension (OR 0.769, P<0.041) were more prevalent in men than in women (OR 0.472, P<0.001; OR 0.902, P<0.673; and OR 0.602, P<0.080, respectively, in the subset).

In univariate analysis, all metabolic and anthropometric traits were significantly associated with NAFLD. In logistic regression analysis, after adjustment for sex, BMI ≥24 kg/m$^2$ (OR 8.494, 95% CI 5.581 to 12.928; P<0.001),

**Table 3** Multivariate adjusted models for liver disease subtypes

| Variables | OR | P value |
|---|---|---|
| (A) Subset | | |
| Sex (male) | 3.484 | <0.001 |
| Obesity | 11.738 | <0.001 |
| Body fat | 2.285 | 0.033 |
| Elevated liver enzyme | 2.237 | 0.105 |
| Hypertension | 1.865 | 0.089 |
| (B) Whole population | | |
| Sex (male) | 2.646 | <0.001 |
| Obesity | 8.494 | <0.001 |
| Body fat | 1.833 | 0.001 |
| Blood lipid | 1.340 | 0.046 |
| Blood glucose | 3.324 | <0.001 |
| Impaired liver function | 1.859 | 0.094 |
| Elevated liver enzyme | 3.150 | <0.001 |
| Hypertension | 1.451 | 0.017 |

Conditions were defined as follows: female sex=1, obesity (body mass index ≥28 kg/m$^2$), hypertension (blood pressure ≥140/90 mm Hg or antihypertensive drug treatment), body fat ratio ≥25 for women or ≥20 for men, blood fasting glucose ≥6.1 mmol/L, and liver enzyme elevation (AST ≥40 U/L or ALT ≥40 U/L).
ORs and the corresponding P values derived from binary or multiple logistic regression analyses using SPSS V.17.0 software.
ALT, alanine aminotransferase; AST, aspartate aminotransferase.

BFR ≥25 for women and ≥20 for men (OR 1.833, 95% CI 1.286 to 2.756; P=0.001), triglycerides ≥1.7 mmol/L (OR 1.340, 95% CI 1.006 to 1.785; P=0.046), fasting glucose ≥6.1 mmol/L (OR 3.324, 95% CI 1.888 to 5.850; P<0.001), BP ≥140/90 mm Hg or antihypertensive drug treatment (OR 1.451, 95% CI 1.069 to 1.970; P=0.017), uric acid ≥357 µmol/L for women and ≥416 µmol/L for men (OR 2.755, 95% CI 2.009 to 3.778; P<0.001), and total OAHOA <5 nmol/L (OR 1.340, 95% CI 1.006 to 1.785; P=0.046) (after adjustment for age ranges and sex in the subset, ORs for BMI, BFR and BP were 11.738 (95% CI 5.193 to 26.530, P<0.001), 2.285 (95% CI 1.067 to 4.893, P=0.033) and 1.865 (95% CI 0.910 to 3.824, P=0.089), respectively) were independent predictors of NAFLD (table 3A,B and table 4). The prevalence of NAFLD demonstrated an increasing trend with lowering sphingosines (OR 1.448, P=0.156; table 4). Substitution of age ranges for continuous data (ie, absolute age) in logistic regression did not modify the relationships. When each metabolic trait was analysed independently, after adjustment for age and sex, ORs increased for obesity and hypertension with increasing age, and ORs increased for BFR with increasing age for patients aged <50 years and decreased for patients aged ≥50 years. However, significant interactions were only observed between age and obesity (P<0.001), and between age and hypertension (P<0.001).

## Association between NAFLD and liver enzymes

Participants with NAFLD had higher liver enzyme levels than participants without this condition (P<0.001).

**Table 4** Metabolites associated with non-alcoholic fatty liver disease

| Metabolite | OR | P value | AUC | P value |
|---|---|---|---|---|
| Uric acid | 2.755 | <0.001 | 0.579 | <0.001 |
| Sphingosine | 1.448 | 0.156 | 0.489 | 0.760 |
| OAHOA | 1.340 | 0.046 | 0.612 | 0.001 |

Conditions were defined as sphingosine <2 nmol/L and OAHOA (oleic acid-hydroxy oleic acid) <5 nmol/L.
ORs, area under curve (AUC) and the corresponding P values derived from binary or multiple logistic regression, or receiver operating characteristic curve analyses using SPSS V.17.0 software.

 Hu X-Y, *et al*. *BMJ Open* 2018;**8**:e019974. doi:10.1136/bmjopen-2017-019974

Abnormal liver enzymes were also significantly associated with NAFLD, independent of sex (OR 3.150, P<0.001). Normal liver enzyme levels, defined according to local guidelines (ALT <40 U/L), were found in 85% of participants with NAFLD.

## DISCUSSION

Our results demonstrated that NAFLD was strongly associated with metabolic traits including higher body fat, obesity, hyperlipidaemia and impaired fasting glucose. We observed a higher prevalence of NAFLD in men than in women. We conformed the association of impaired uric acid and abnormal liver enzymes with NAFLD, and further identified impaired total OAHOA as a novel biomarker of NAFLD prevalence.

The average and peak prevalence of NAFLD in this study were 24.7% and more than 26%, respectively, close to the global prevalence of NAFLD, which was 25.24%, with the highest prevalence in the Middle East (31.79%) and South America (30.45%), and the lowest in Africa (13.48%).[27] The average and peak prevalence of NAFLD in Shanghai[5] were 20.82% and 28.44%, respectively, and those in Guangdong[6] were 17.2% and 27.4%, respectively. The former study was conducted from 2002 to 2003, and the latter in 2005. Our study, performed in 2010, revealed that the average prevalence is increasing, and the peak prevalence occurred at a younger age (30–40 years) than was reported by the previous studies in Shanghai (age of 60–69),[5] Guangdong (age of 60–69)[6] and worldwide (mean age of 70–79).[27] Thus, more attention should be given to the health of people in their 30s, when many marry and adopt lifestyle changes, especially in urban China. Unfortunately, we did not have access to information on participants' activity levels, and thus the effects of physical activity on NAFLD were not assessed. The reasons underlying the observed earlier peak prevalence require further investigation.

We observed that male sex was a risk factor for NAFLD. This finding differs from the results of the Guangdong study[6] and the Rotterdam study.[13] However, this finding is similar to that of the Shanghai study.

There are several possibilities for this discrepancy. First, the observed sex difference in NAFLD prevalence may be the result of body fat, which is a risk factor for NAFLD and was higher in men than in women in this study. Therefore, male sex may promote NAFLD through mediators such as increased body fat and hormonal change. Second, the sex difference in NAFLD may also result from a lower prevalence of boosted serum triglycerides, glucose and hypertension in women compared with men. A lower prevalence of boosted serum triglycerides, glucose and hypertension in women was reported in a previous study.[28] However, a causal relationship between NAFLD and these serum metabolic markers, as suggested by previous research,[29 30] could not be established in our research due to its cross-sectional characteristics. Third, abnormal liver enzymes were more prevalent in men than in women, and both were independent predictors of NAFLD.

We observed that the associations of the identified metabolites with NAFLD risk were pronounced in the Wuhan cohort. These metabolites might play a major role in the development of onset NAFLD. Uric acid was particularly associated with NAFLD in men with type 2 diabetes, besides insulin resistance and other metabolic factors.[31] High concentrations of uric acid induce the accumulation of reactive oxygen species in hepatocyte mitochondria, ultimately leading to mitochondrial damage,[32] and uric acid was positively correlated with other metabolic traits and was a risk factor for type 2 diabetes mellitus with NAFLD.[33]

Serum uric acid (SUA) is produced by purine nucleotide catabolism.[34] SUA subjects have an about twofold higher risk of NAFLD, as shown by ultrasonography. The risk may be independent of age, sex and obesity (indicated by BMI and waist circumference). SUA is an independent risk factor in NAFLD in both Uyghurs and Hans in north-western China.[35] Among participants with elevated SUA levels, women likely showed a greater risk of NAFLD than men.[36]

OAHOAs are present in humans, which are an endogenous branched FAHFAs, and levels are reduced with obesity and insulin resistance. OAHOA levels in serum are correlated highly with whole-body insulin sensitivity. OAHOAs are endogenous GPR120 ligands and may also exert anti-inflammatory effects in vivo through fatty acid-activated G-protein-coupled receptors (GPCRs), such as GPR120. Androgen-induced protein 1 (AIG1) and androgen-dependent TFPI-regulating protein (ADTRP), atypical integral membrane hydrolases, degrade bioactive FAHFAs,[37] and branched FAHFAs are preferred substrates of MODY8, a protein carboxyl ester lipase.[38] Changes in the concentrations of these anti-inflammatory metabolites and in their signalling networks may provide new targets for metabolic and inflammatory diseases.[22] GLUT4 expression and levels of FAHFAs with antidiabetic and anti-inflammatory effects regulate de novo lipogenesis in adipocytes.[39] Compared with healthy controls, FAHFAs are significantly decreased in the sera of patients with breast cancer.[40] Branched FAHFAs protect against colitis by controlling gut innate and adaptive immune responses.[41]

Our study also has some potential limitations. First, we cannot rule out the possibility of biased selection, since non-responders may have had different morbidities. Volunteers in research studies tend to be better educated, healthier and have better lifestyles.[42] Consequently, estimations of the prevalence of NAFLD may have been biased. Second, only 447 of the 1479 participants provided information on their ages, and we did not obtain data on participants' educational levels or incomes. Third, specific OAHOA isomers[22] require further study.

## CONCLUSIONS

NAFLD is common in the study population. In this observational cross-sectional primary survey, we found that NAFLD was associated with male sex, and the peak prevalence occurred at a younger age. In particular, NAFLD was correlated with uric acid. Moreover, we identified OAHOA as a novel biomarker of NAFLD. Further studies are needed to explore the potential factors contributing to these relationships. However, this study is valuable in that it provides a reference on the prevention of NAFLD and related metabolic diseases with the rapid urbanisation, technological advancement and population ageing in China.

**Contributors** Designed the study: X-JZ. Analysed the data: X-YH, YL, L-QL, YZ, J-HL, S-CH, X-JZ. Wrote and critically reviewed the manuscript: X-YH, WZ, LL, LZ, ZL, X-JZ.

**Funding** This work was supported in part by the Opening Project of Hubei Key Laboratory of Lipid Chemistry and Nutrition of Oil (201506) and the National Nature Science Foundation of China (21602166).

**Disclaimer** The Opening Project of Hubei Key Laboratory of Lipid Chemistry and Nutrition of Oil and the National Nature Science Foundation of China had no role in this study.

**Competing interests** None declared.

**Patient consent** Obtained.

**Ethics approval** The ethics committee of Wuhan Union Hospital approved the study, and all participants provided written informed consent prior to enrolment.

**Provenance and peer review** Not commissioned; externally peer reviewed.

**Data sharing statement** Data are available on request.

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
