## [Reviewer comments · BMJ Open]

ARTICLE DETAILS

TITLE (PROVISIONAL)	Risk Factors and Biomarkers of Non-alcoholic Fatty Liver Disease: an observational cross-sectional population survey
AUTHORS	Hu, Xiao-Yu; Li, Yun; Li, Long-Quan; Zheng, Yuan; Lv, Jia-Hong; Huang, Shu-Chun; Zhang, Wei-Nong; Liu, Liang; Zhao, Ling; Liu, Zhiguo; Zhao, Xiu-Ju

VERSION 1 – REVIEW

REVIEWER	Masaru Baba Japan Community Health Care Organization Hokkaido Hospital, Japan
REVIEW RETURNED	06-Nov-2017

GENERAL COMMENTS	Comments to the Authors This manuscript has described about the risk factors and biomarkers of NAFLD. The authors described the prevalence of NAFLD and correlated factors of NAFLD in this study. And the authors identified that OAHOA was a noble biomarker of NAFLD. This seems to be an article of limited interest to the researchers on the same field, but should be modified and revised as follows, Major problems 1. The authors should discuss more about the similarities and differences among the previously published results and this study results. There are many publications showing the prevalence and the associated factors about NAFLD. For Example, Z. Younossi et al (Hepatology. 2016; 64: 73-84) reported the global prevalence, incidence, progression, and outcomes of NAFLD. Please take these into account in the Discussion section.2. The authors should explain more about the oleic acid-hydroxy oleic acid, especially the role of the oleic acid-hydroxy oleic acid in NAFLD patients.3. The authors should discuss more about the relationship between metabolic diseases and the urbanization, technological advancement, and population aging. In the Younossi's paper (Hepatology. 2016; 64: 73-84), the relatively high prevalence of NAFLD found in the Asian population, as well as their higher prevalence of obesity. Other regions like North America or Europe are the same or more urbanize and have the same or more technological advancement.4. The authors should describe the NAFLD diagnosis criteria in this study. And the authors should indicate the alcohol consumptions in this study populations.5. The authors described that obesity was defined as BMI more than 24 kg/m² in this paper. The authors should describe the
--

	reasons of the authors criteria. In WHO criteria, a person with a BMI of 30 or more is generally considered obese, and a person with a BMI equal to or more than 25 is considered overweight. Minor problem 1) Needs some language and sentence corrections before revision. Discretionary Revisions My opinion is that the article should be modified in order to consider OAHOA as a noble marker of NAFLD prevalence. The prevalence and the associated factors of NAFLD only should be established.
--	--

REVIEWER	Vasilios G. ATHYROS, MD Aristotelian University, Thessaloniki, Greece
REVIEW RETURNED	18-Nov-2017

GENERAL COMMENTS	The review paper bmjopen-2017-019974 by Dr Xiao-Yu Hu et al, entitled "Risk Factors and Biomarkers of Non-alcoholic Fatty Liver Disease" has relevance to the audience of BMJ-OPEN. The aim of this paper was to determine risk factors and biomarkers of non-alcoholic fatty liver disease (NAFLD). Authors conclude that NAFLD was common among the study participants. In particular, NAFLD was correlated with uric acid. We identified oleic acid-hydroxy oleic acid (OAHOA) as a novel marker of NAFLD prevalence. It provides a reference on the prevention of NAFLD and related metabolic diseases with the increasing urbanization, technological advancement, and population aging in China. Major Comments for the authors  1. This is a retrospective study with collection of data 7 years ago in the triple Wuhan city (10 mil people): "Our study, performed in 2010, revealed that the average prevalence is increasing, and the peak prevalence occurred at a younger age than was reported by the previous studies in Shanghai (age of 60-69) [5] and Guangdong (age of 60-69)". It is not clear if different populations of different ages and at different time periods can be compared. 2. It is very useful that this study verifies the important role of uric acid as a NAFLD risk factor and reports OAHOA as a novel NAFLD risk factor. 3. The risk factors reported are well known for many years now (more than 12,000 studies in PubMed on NAFLD), mainly in western countries. During the last few years more than 700 papers were performed in Asia. If we need data (which are not so recent) for China on NAFLD risk factors this study could be suitable for this Journal, otherwise not. 4. There is no mention of non-alcoholic steatohepatitis (NASH), of hepatocellular carcinoma, of cardiovascular disease risk (the main reason of death in NASH patients), and finally no mention of treatment of NAFLD (not even the words of the above is mentioned in the text). This could be done after 7 years of the initial data
--

	collection. Thus, the results of this review study cannot have any practical implications. People, all over the world, do not change their lifestyle for long. 5. The references are not up to date.
--	--

VERSION 1 – AUTHOR RESPONSE

Reviewer(s)' Comments to Author:

Reviewer: 1

Reviewer Name: Masaru Baba

Institution and Country: Japan Community Health Care Organization Hokkaido Hospital, Japan

Please state any competing interests: None declared

Please leave your comments for the authors below

Comments to the Authors

This manuscript has described about the risk factors and biomarkers of NAFLD. The authors described the prevalence of NAFLD and correlated factors of NAFLD in this study. And the authors identified that OAHOA was a noble biomarker of NAFLD.

This seems to be an article of limited interest to the researchers on the same field, but should be modified and revised as follows,

We thank the reviewer and have revised the manuscript carefully.

Major problems

1. The authors should discuss more about the similarities and differences among the previously published results and this study results. There are many publications showing the prevalence and the associated factors about NAFLD. For Example, Z. Younossi et al (Hepatology. 2016; 64: 73-84) reported the global prevalence, incidence, progression, and outcomes of NAFLD. Please take these into account in the Discussion section.

We thank the reviewer and have revised the Discussion section thoroughly.

2. The authors should explain more about the oleic acid-hydroxy oleic acid, especially the role of the oleic acid-hydroxy oleic acid in NAFLD patients.

We have explained more about the role of OAHOA in patients.

3. The authors should discuss more about the relationship between metabolic diseases and the urbanization, technological advancement, and population aging. In the Younossi's paper (Hepatology. 2016; 64: 73-84), the relatively high prevalence of NAFLD found in the Asian population, as well as their higher prevalence of obesity. Other regions like North America or Europe are the same or more urbanize and have the same or more technological advancement.

China has urbanization, technological advancement, and population aging in a short term (around 40 years), which may be or may not be the same as the situation of Japan, Europe and North America in a relatively long term (around 200-300 years). In the Younossi's paper (Hepatology. 2016; 64: 73-84), the peak prevalence is in mean age 70-79; whilst, our study revealed that the peak prevalence occurred at a younger age (30-40 years).

4. The authors should describe the NAFLD diagnosis criteria in this study. And the authors should indicate the alcohol consumptions in this study populations.

The NAFLD diagnosis criteria and the alcohol consumptions are in the section DIAGNOSIS OF NAFLD: "NAFLD was diagnosed by the technician according to ultrasonography" and "Individuals ... were excluded from the analyses: (1) excessive alcohol consumption "

5. The authors described that obesity was defined as BMI more than 24 kg/m² in this paper. The authors should describe the reasons of the authors criteria. In WHO criteria, a person with a BMI of 30 or more is generally considered obese, and a person with a BMI equal to or more than 25 is considered overweight.

It is a typo error and obesity was defined as BMI more than 28 kg/m², based on latest survey in China.

Minor problem

1) Needs some language and sentence corrections before revision.

Elsevier Webshop has provided writing assistance for the main body of our manuscript.

Discretionary Revisions

My opinion is that the article should be modified in order to consider OAHOA as a noble marker of NAFLD prevalence. The prevalence and the associated factors of NAFLD only should be established. We described the prevalence and correlated factors of NAFLD in this study. And we identified OAHOA as a biomarker of NAFLD preliminarily and the role of OAHOA should be verified in a multi-center perspective cohort study.

Reviewer: 2

Reviewer Name: Vasilios G. ATHYROS, MD

Institution and Country: Aristotelian University, Thessaloniki, Greece

Please state any competing interests: "None declared"

Please leave your comments for the authors below

The review paper bmjopen-2017-019974 by Dr Xiao-Yu Hu et al, entitled "Risk Factors and Biomarkers of Non-alcoholic Fatty Liver Disease" has relevance to the audience of BMJ-OPEN.

The aim of this paper was to determine risk factors and biomarkers of non-alcoholic fatty liver disease (NAFLD).

Authors conclude that NAFLD was common among the study participants. In particular, NAFLD was correlated with uric acid. We identified oleic acid-hydroxy oleic acid (OAHOA) as a novel marker of NAFLD prevalence. It provides a reference on the prevention of NAFLD and related metabolic diseases with the increasing urbanization, technological advancement, and population aging in China. We thank the reviewer.

Major Comments for the authors

1. This is a retrospective study with collection of data 7 years ago in the triple Wuhan city (10 mil people): "Our study, performed in 2010, revealed that the average prevalence is increasing, and the peak prevalence occurred at a younger age than was reported by the previous studies in Shanghai (age of 60-69) [5] and Guangdong (age of 60-69)". It is not clear if different populations of different ages and at different time periods can be compared.

In China, the social-economical situations of Wuhan, Shanghai and Guangdong are similar, and we have surveyed for observing possible similarity or difference across different time periods.

2. It is very useful that this study verifies the important role of uric acid as a NAFLD risk factor and reports OAHOA as a novel NAFLD risk factor.

We thank the reviewer.

3. The risk factors reported are well known for many years now (more than 12,000 studies in PubMed on NAFLD), mainly in western countries. During the last few years more than 700 papers were performed in Asia. If we need data (which are not so recent) for China on NAFLD risk factors this study could be suitable for this Journal, otherwise not.

China has urbanization, technological advancement, and population aging in a short term (around 40 years), which may be or may not be the same as the situation of Japan, Europe and North America in a relatively long term (around 200-300 years). Thus we have carried out this study.

4. There is no mention of non-alcoholic steatohepatitis (NASH), of hepatocellular carcinoma, of cardiovascular disease risk (the main reason of death in NASH patients), and finally no mention of treatment of NAFLD (not even the words of the above is mentioned in the text). This could be done

after 7 years of the initial data collection. Thus, the results of this review study cannot have any practical implications. People, all over the world, do not change their lifestyle for long. The study participants have no HCC or cardiovascular disease; we have added this information. Frankly speaking, this is a preliminary observational cross-sectional population survey and the information on treatment of NAFLD is not our aim.

5. The references are not up to date.

We thank the reviewer and the references are updated.

VERSION 2 – REVIEW

REVIEWER	Vasili9os G. ATHYROS, MD Aristotle University of Thessaloniki, Greece
REVIEW RETURNED	09-Jan-2018
GENERAL COMMENTS	Any issues of the original paper were resolved in the revision.

VERSION 2 – AUTHOR RESPONSE

Editors comments:

- We noted that you have included a new author (Chaodong Wu) in the author list of your revised manuscript, but have not added this additional author to the submission system. When submitted your revision please add the new author into the submission system.

We have removed the author Chaodong Wu in our manuscript.

- We noted that you are still including a CONSORT checklist alongside your revised manuscript. Please remove the CONSORT checklist, which is for reporting of clinical trials and please include a copy of the STROBE checklist for reporting of observational studies, indicating the page/line numbers of your manuscript where the relevant information can be found (<https://strobe-statement.org/index.php?id=strobe-home>).

The STROBE checklist has been included.

- Please remove the term 'large-scale' from the title of your manuscript.

We have removed it from the title and the main body of our manuscript.

- Please modify the 'Main outcome measures' section of your abstract to include details of the risk factors and metabolic biomarkers measured.

We have added the details.

- Please clarify the sample size of your study on which your analysis has been carried out as this is reported inconsistently throughout your manuscript. You state throughout your manuscript that your sample is 1500. However, in the methods section you state that "a total of 1479 study participants were included in the final analysis", in which case your sample is 1479, and not 1500. However, in the

results section of your abstract you present results from only 454 participants; "Data from the 454 participants". Furthermore, in the results section of your manuscript you state that analysis has been carried out on data from 447 participants who "reported their age and had no more than one disease, if any (denoted as the "subset")". Firstly, please clarify if the 'subset' consists of 454 or 447 participants and modify your manuscript accordingly. Secondly, please modify your manuscript throughout to make it clear which analysis has been carried out using data from your full sample (1479) and which analysis has been carried out using data only from the 'subset'.

We have modified our sample from 1500 to 1479 and the subset unified to 447. The corresponding table has modified accordingly.

- Please improve the reporting of the statistics throughout your manuscript. For example, please include the 95% CI for the reporting of all OR including those in the abstract.

We thank the editors and have improved statistics. Due to abstract limit 300 WORDS, and added details in the 'Main outcome measures' section, the 95% CI for the reporting of all OR in the abstract is not included online.

Reviewer(s)' Comments to Author:

Reviewer: 2

Reviewer Name: Vasilios G. ATHYROS, MD

Institution and Country: Aristotle University of Thessaloniki, Greece

Please state any competing interests or state 'None declared': None declared

Please leave your comments for the authors below

Any issues of the original paper were resolved in the revision.

We thank this reviewer.